# Assessment of Surface Water Quality in the Podu Iloaiei Dam Lake (North-Eastern Romania): Potential Implications for Aquaculture Activities in the Area

Cornelia Amarandei [1,2], Alina-Giorgiana Negru [2], Laurentiu-Valentin Soroaga [1,2], Simona-Maria Cucu-Man [1], Romeo-Iulian Olariu [1,2] and Cecilia Arsene [1,2,*]

1 Department of Chemistry, Faculty of Chemistry, "Alexandru Ioan Cuza" University of Iasi, 11 Carol I, 700506 Iasi, Romania; cornelia.amarandei@uaic.ro (C.A.); laurentiu.soroaga@uaic.ro (L.-V.S.); sman@uaic.ro (S.-M.C.-M.); oromeo@uaic.ro (R.-I.O.)

2 Integrated Centre of Environmental Science Studies in the North Eastern Region, "Alexandru Ioan Cuza" University of Iasi, 11 Carol I, 700506 Iasi, Romania; alina.negru@uaic.ro

* Correspondence: carsene@uaic.ro; Tel.: +40-232-201354; Fax: +40-232-201313

**Abstract:** The Podu Iloaiei Dam Lake located on the Bahluet River from Bahlui hydrographic basin, north-eastern Romania, is one of the most important water resources used for aquaculture activities in the region of interest. In the present study, the chemical composition related to water-soluble ions and elements was assessed in both water and sediment samples collected from the area of interest during July 2017 and October 2017, representative months for warm and cold seasons, respectively. Water-soluble ions ($H_3C_2O_2^-$, $HCO_2^-$, $C_2O_4^{2-}$, $F^-$, $Cl^-$, $NO_2^-$, $Br^-$, $NO_3^-$, $SO_4^{2-}$, $Li^+$, $Na^+$, $NH_4^+$, $K^+$, and $Ca^{2+}$) were analyzed by ion chromatography, while inductively coupled plasma mass spectrometry was used to quantify water-soluble fractions of elements (Be, B, Mg, Al, Ti, V, Cr, Mn, Fe, Co, Ni, Cu, Zn, Ga, Ge, Rb, Sr, Mo, Ru, Pd, Ag, Cd, Sn, Sb, Te, Ba, Ir, Tl, Pb, Bi, and U). Evidence was obtained on the contributions of both anthropogenic and natural (pedologic) related sources in controlling the chemical composition of the water and sediment samples in the area. Analysis of Piper diagrams revealed the existence of $CO_3^{2-}/HCO_3^-$ and $Ca^{2+}/Mg^{2+}$ as dominant species for the sediment samples. The interest water pool was found to be oligotrophic over the warm period and eutrophic over the cold period. Overall, abundances and the association of chemical species in the area seemed to be controlled by a complex interplay between the water body's main characteristics, meteorological factors, and anthropogenic activities. Moreover, the present results suggest that precautions should be taken for physicochemical parameter monitoring and prevention acts for surface water quality assurance in order to control the potential negative influence of some chemical parameters on fish productivity. Reported data also have a high potential to be used by experts in the field of developing lake water management policies for a sustainable exploitation of various aquatic systems.

**Keywords:** heavy metals; water-soluble ions; spatial distribution; nutrients; Podu Iloaiei Dam Lake





## 1. Introduction

Changes in surface water's quality due to dramatically increased chemicals and nutrient materials threaten aquatic ecosystems and the environmental conditions. Evaluation of water and sediment physicochemical characteristics is highly important for quality control of surface water resources and for identification of potential exploitation. Safe water for fish farming requires water parameters to be in certain ranges [1], with some (e.g., pH, water hardness, etc.) being more difficult to be controlled in natural water reservoirs. Additionally, heavy metal contamination of the aquatic environment has drawn special attention because of the persistence, bioaccumulation, and potential adverse effects on biota [2]. Understanding the physicochemical parameters and evaluation of the anthropogenic impact on surface water and sediment are highly important for water quality

assessments and aquatic environmental protection. The sediment is a major sink of heavy metals due to scavenging of metals from the water column. However, it can also be a potential source of metal contamination into the water column and aquatic biota due to remobilization processes. The behavior of heavy metals in sediments, and therefore their distribution, is affected by changes in sediment grain size, organic content, and the redox regime [3]. The construction of dams on rivers and the loss of water discharge could cause the accumulation of anthropogenic waste in water reservoirs [4].

The water used for aquaculture purposes requires appropriate conditions to ensure fishes' growth (fish seeds, feed, fertilizers), and, along with the fish farming activities, chemical residues and microorganisms are usually produced as wastes [5]. Fertilizers are highly soluble substances frequently used for aquaculture purposes [6]. Most of these chemicals are well known for their potential to release nutrients that can cause eutrophication of natural waters. In aquaculture there are also other substances that are less frequently used. Among them, liming material, oxidants, disinfectants, osmoregulators, algicides, coagulants, herbicides, and probiotics should be mentioned [6]. Moreover, in aquaculture the ionic composition of water is important for fish hatching, feeding, development, larval growth, and survival [7]. The concentrations of $Ca^{2+}$ and $Mg^{2+}$ divalent cations play a very important role in the ionic regulation of freshwater fish, with both ions modulating the branchial permeability [8]. Water contamination with agricultural, industrial, or human waste could result in aquaculture product contamination and food safety concerns. However, nowadays there is an increased concern over the potential harm of aquaculture effluents on receiving water bodies and worries over the contamination of aquatic food products with bioaccumulative and potentially harmful chemicals [5,9]. At the European scale, waters' status and pressures are presented in Report No. 7/2018 of the European Environment Agency (EEA) [10]. The ecological status and potential of surface water bodies are assessed based on biological, physicochemical (nutrients, organic pollution, acidification), and hydromorphological elements. In terms of water quality, five classes, including high, good, moderate, poor, and bad, are usually referred to. When speaking about the chemical status derived from the evaluation of priority substances, waters can be assigned as "good" or "failing to achieve good." The EEA report indicates that, at the European scale, 38% of surface water bodies are in good chemical status, 46% are under the critical thresholds for this status, and 16% have an unknown status. Moreover, at the European level, just 40% of the surface water bodies are in good or high ecological status or potential.

The Podu Iloaiei water reservoir, located on the Bahluet River from the Bahlui River basin in north-eastern Romania, is one of the most important water resources used for aquaculture activities in the region of interest. It has both economic and socio-politic value since the area is considered as one of the most anthropized hydrological basins in the country. Anthropogenic activities have important implications for the formation and evolution of water resources in the Bahlui hydrological basin. Waste from the municipalities and farms located along the Bahluet River, agricultural runoff, the application of chemical fertilizers in agriculture, and transport routes around the reservoir are the potential pollution sources in the investigated area [11]. Nevertheless, little is known about the chemical parameter levels in the water and sediment of Podu Iloaiei Dam Lake.

Over the years, Directive 2000/60/EC [12] of the European Parliament and of the council establishing the framework for community action in the field of water policy have been implemented in Romania through the Ministry of Waters and Environmental Protection Order No. 161/2006 [13]. According to the norms for classification of surface water quality, the ecological status of water bodies is established on the base of five quality classes for surface waters (I–V) (Table S1 of the Supplementary Material).

For the Bahlui River basin, in terms of required parameters for establishing water quality class, only very few reports can be identified in the literature [14,15]. Both previous studies report measured values for chemical parameters limited to pH, $CBO_5$, CCO-Mn, $NH_4^+$, $NO_2^-$, $NO_3^-$, $PO_4^{3-}$, and $Mg^{2+}$ [14,15]. There is also one study reporting about

the anthropogenic impact on urban soil properties from the Bahlui Floodplain [16]. The treatment station of the town of Targu Frumos and some livestock farms located along the Bahluet River are suggested as the main pollution sources of Podu Iloaiei Dam Lake [14]. The impact of anthropogenic activities on water quality parameters of glacial lakes from Rodnei mountains was also reported in a more recent study [17]. Relatively high concentrations of $NH_4^+$, $NO_2^-$, Ca, Mg, Fe, and Cu were identified even in this water representative for a more pristine environment.

Reports on the photochemistry induced either by hydroxyl or nitrite radicals in natural water samples collected from wells in the Letcani area (north-eastern Romania, Bahlui River basin) [18–21] or from selected Romanian lakes [22] are present in the literature. In some studies, the authors also report on experimental evidence for aromatic nitration upon irradiation of natural water samples since, in the interest zones, very high nitrate levels were identified [20,22,23]. However, other representative studies in Europe highlight fairly well the importance of surface water investigations [24].

In Romania, the governmental summary report from 2018 of the National Agency for Environmental Protection classifies the Podu Iloaiei Dam Lake water as moderate (class III) in terms of general physicochemical parameters, but no results are presented for specific pollutants and the chemical status [25]. In the present work, accurate concentrations of water-soluble ions and water-soluble elements in water and sediment samples from Podu Iloaiei Dam Lake were reported. The main objectives of the performed investigations were: (1) to determine the level of heavy metal and ion concentrations for evaluation of water quality status by seasonal change, (2) to investigate the spatial distributions of chemical parameters in water and sediment, (3) to identify their potential sources using the multivariate statistical analysis, and (4) to indicate the suitability of water quality for intensive fish farming purposes. Moreover, the work reported in the present study may be considered as a key reference point for further investigation on potential degradation processes of photolabile organic compounds or in identifying the role played by some elements, e.g., Fe, in the photoproduction of OH radicals under specific conditions.

## 2. Materials and Methods

### 2.1. Study Area and Sampling Strategy

The Podu Iloaiei Dam Lake is located in the north-eastern part of Romania (Figure 1) on the Bahluet River at a distance of 2.5 km upstream of the confluence with the Bahlui River, at 400 m upstream from the town of Podu Iloaiei, Iasi County. The Bahluet River is an affluent of the Bahlui River located in the area between the Moldavian Plain and the Central Moldavian Plateau. The average altitude of the hydrological basin is 159 m above sea level, and the multiannual average flow rate of Bahluet River is 1.06 m s$^{-1}$ [26]. The main water input for Podu Iloaiei Dam Lake is Bahluiet River, and other sources include snow melting and natural springs. The Bahluiet River basin has a total surface of 550 km$^2$, a length of 50 km, an average slope of 13%, and a sinuosity coefficient of 1.23 [27].

The Podu Iloaiei Dam Lake was made to regulate the Bahluet River water flow rate, and it is used for fish farming and irrigation. Podu Iloaiei Dam Lake has a water surface of 251 ha and a dam of 14.1 m in elevation and 640 m in width [28]. The area and volume of Podu Iloaie Lake can vary considerably throughout the year. The water level is usually lower during the summer (June, July, and August) and higher during major rainfall events in the autumn (September, October, and November). The lake water is evacuated for the winter time.

The hydrographic basin is characterized by alluvial soils, carbonated alluviums, soils rich in sodium sulphates, meadow chernozem (formed on clayey and maroon substrates), and gray soils. Surface deposits are represented by monotonous succession of marches, clays, and sands with some intercalations of limestone sandstone, which is disposed monoclinally [11]. Bessarabia deposits (clay marls with intrusions of sand) from the Sarmatian age dominate the basin. Quaternary deposits from the Pleistocene age and Holocene deposits are also located in the Bahluet River area [27].

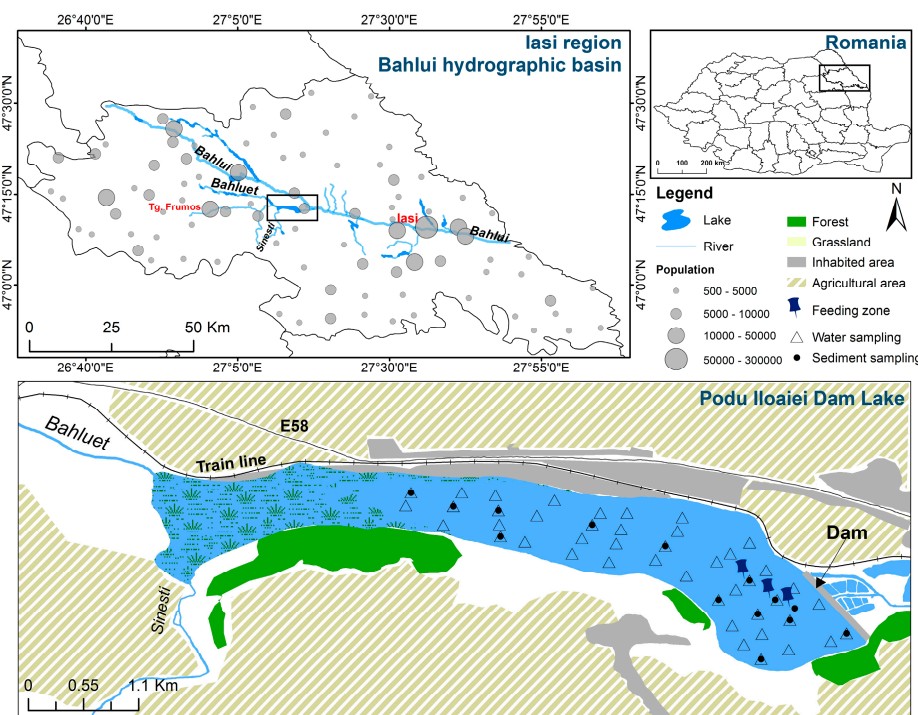

**Figure 1.** Location of the study area, water, and sediment sampling points.

The climate of the area is temperate–continental, with warm summers and cold winters. During the year, the temperature typically ranges from −5 °C to 28 °C, rarely reaching below −14 °C or above 33 °C, and the annual rainfall average is 589 mm. In the year the sampling was undertaken, May and June 2017 recorded the highest average precipitation values of 44 and 54 mm, respectively, and October 2017 was the month with the lowest rainfall, with an average of 27 mm.

According to the National Institute of Statistics, in the Prut–Barlad hydrographic area, including the Bahlui hydrographic area, about 71% of the surface is used for agriculture, 14% is covered by forests, 12% is occupied by construction, and 3% is occupied by surface waters [29]. The Podu Iloaiei Dam Lake is used for fish farming and is an important source of water for fishponds downstream of the dam [26].

In order to evaluate the chemical parameters of the investigated environment, water and sediment samples were collected from Podu Iloaiei Dam Lake in 50 mL polypropylene bottles in two sessions. The entire set of samples comprised 88 water samples and 28 sediment samples collected on 16 July 2017 and 5 October 2017, respectively. The sampling strategy was designed so as to obtain a significant number of data points for an accurate statistical treatment or for high-resolution distribution maps. The availability of technical resources was the limitative factor in increasing the sampling frequency or the number of the samples for each matrix. The distribution of the water and sediment sampling points is presented in Figure 1.

Water samples were collected from a ~10 cm depth (from the water surface layer), and sediment samples were collected from a ~20 cm depth (from the water–soil interface). The samples were stored at −4 °C until the analysis, which was performed within a maximum of 72 h after sampling.

### 2.2. Materials and Methods

2.2.1. Water-Soluble Ions Analysis by Ion Chromatography

The following inorganic anions: chloride ($Cl^-$), fluoride ($F^-$), bromide ($Br^-$), nitrite ($NO_2^-$), nitrate ($NO_3^-$), phosphate ($PO_4^{3-}$) and sulphate ($SO_4^{2-}$); organic anions: acetate ($H_3C_2O_2^-$), formate ($HCO_2^-$), and oxalate ($C_2O_4^{2-}$); and inorganic cations: sodium ($Na^+$),

potassium ($K^+$), lithium ($Li^+$), ammonium ($NH_4^+$), magnesium ($Mg^{2+}$), and calcium ($Ca^{2+}$) were quantified by ion chromatography.

The sediment samples were dried at 50 °C for 72 h in a ventilated oven (Ecocell), and 0.1–0.2 g of dry sediments were weighted and subjected to extraction with deionized water by ultrasound-assisted extraction (ultrasonic bath Elmasonic S70H, Elma, Singen, Germany) for 45 min at room temperature. The water samples and the aqueous extract of the sediment samples were filtered before analysis using cellulose acetate filters (ADVANTEC) with a pore diameter of 0.20 μm.

The ion determination was performed by the DIONEX ICS-5000+ DP system (Thermo Scientific, Waltham, MA, USA). In the case of anion determination, a 4.5/1.4 mM $CO_3^{2-}$/$HCO_3^-$ solution was used for elution at a flow rate of 1.2 mL min$^{-1}$, using an AS22 column (4 × 250 mm) and an AERS 500 suppressor (4 mm). A 20 mM methanesulfonic acid solution was used as mobile phase at a flow rate of 1.0 mL min$^{-1}$ for cation separation. In this case, a CS12A column (4 × 250 mm) and a CSRS 300 suppressor (4 mm) were used.

The external standard method was used to quantify water-soluble ions in both water and sediment samples. Calibration curves were performed in the concentration range of 50–3000 μg L$^{-1}$ for anions and 50–2500 μg L$^{-1}$ for cations. For the calibration curves, the required standard solutions were prepared by appropriate dilutions of seven anions-II and six cations-II Dionex standards. Calibration curves of water-soluble organic anions (acetate, formate, and oxalate) were acquired using solutions prepared with sodium salts of the anions (sodium acetate, ≥99%, from Merck, Darmstadt, Germany; sodium formate, 99%, and sodium oxalate, 99.9%, from AnalaR NORMAPUR, Leuven, Belgium).

The water samples' dilutions, the extraction of soluble ions from the sediment samples, and the preparation of the mobile phases and standard solutions were performed with deionized water (resistivity of 18.2 MΩ·cm) provided by the Milli-Q Advantage A10 system (Millipore, Burlington, MA, USA).

$HCO_3^-$ is one of the major anions present in the water samples but cannot be determined by ion chromatography while using $CO_3^{2-}$/$HCO_3^-$ as eluent. In an attempt to supplement the ionic balance, its estimation was used according to the method proposed by Arsene and co-workers [30]. The statistically significant correlation between the sum of the $Ca^{2+}$ and $Mg^{2+}$ concentrations and the anion deficit estimated from the ionic balance were taken into account. For both the sampling sessions, the linear regressions obtained between these variables are shown in Figure S1 of the Supplementary Material (SM).

### 2.2.2. Water-Soluble Elements Analyses by Inductively Coupled Plasma Mass Spectrometry

The analysis of water-soluble fractions of the elements in water and sediment samples was performed for Be, B, Mg, Al, Ti, V, Cr, Mn, Fe, Co, Ni, Cu, Zn, Ga, Ge, Rb, Sr, Mo, Ru, Pd, Ag, Cd, Sn, Sb, Te, Ba, Ir, Tl, Pb, Bi, and U by inductively coupled plasma mass spectrometry (ICP-MS) using the ICP-MS 7700X system (Agilent Technologies, Santa Clara, CA, USA). For the analysis, a 1 mL water sample and a 1 mL sediment aqueous extract (solution used for the ion chromatography analysis) were diluted with 8.5 mL 3% $HNO_3$. As an internal standard, 0.5 mL of indium (In) solution (1 mg L$^{-1}$) was added in each sample. Internal standard calibration curves were achieved using the multi-element standard solutions with 30 elements (Li, Be, B, Na, Mg, Al, K, Ca, V, Cr, Mn, Fe, Co, Ni, Cu, Zn, Ga, As, Se, Rb, Sr, Mo, Ag, Cd, Te, Ba, Tl, Pb, Bi, and U) and 8 elements (Ti, Ge, Mo, Ru, Pd, Sn, Sb, and Ir). Suprapur® nitric acid (65%) and all standard solutions used for ICP-MS analysis were purchased from Merck, Darmstadt, Germany.

During the analysis, in order to prevent the possible contributions brought by the contaminations, special emphasis was placed on carefully controlling the entire array of analysis steps. During the analysis sequence, for the quality control assurance, blanks and control samples were frequently analyzed. Tables S2 and S3 in the Supplementary Materials show the limits of detection (LoD) and the limits of quantification (LoQ) for the chemical species investigated in the present study.

*2.3. Statistical Analysis of the Data*

The mean, the standard deviation, and the median of the water-soluble ions and elements concentrations were calculated to obtain the variation between two different sampling sessions.

Multivariate statistical tools were applied in Origin 2018 to analyze the associations among different chemical species. Pearson's correlation coefficient and hierarchical cluster analysis were used for source apportionment. The Kolmogorov–Smirnov test was performed to check possible differences between the sampling sessions. Significant differences between the ions and the elements concentrations in both sampling sessions were identified with the value of $p < 0.05$ (95% confidence level).

Cluster analysis of the data with the Ward calculation method [31] was applied to identify possible common sources for the chemical species quantified in the water and sediment samples for both sampling sessions. This is a method of assessing the distance between two clusters by minimizing intra-cluster variability. Other studies in the field of water quality using the Ward method for cluster analysis have been identified in the literature [32,33]. Following the data processing, the graphs known as dendrograms show how the elements are grouped into classes according to the similarities between them.

The geostatistical method was implemented to comprehensively describe the variation of chemical species concentration in the surface water and sediment of Podu Iloaiei Dam Lake. In addition to the cluster analysis, the analysis of the geospatial distributions of the quantified species was used in order to discriminate between the potential contributions of the sources of pedological, atmospheric, or anthropogenic origin. In the literature, geostatistical data processing and the multivariate statistic data evaluation approach have been used for the investigation of pollutants in soil, sediment, and natural waters [34,35]. In order to achieve geospatial distributions of water-soluble ionic species quantified in water and sediment samples, the Google Earth Pro and ArcMap programs included in the ArcGIS 10.2 package were used. Kriging Ordinary interpolation methods with a spherical semivariogram and inverse distance weighting (IDW) interpolation were used for data processing. Compared to IDW, the Kriging method uses the theory of regionalized variables, which assumes that the direction and distance between data reflect a spatial correlation that is applied in explaining their variation by determining the semivariogram [36,37]. For this reason, the Kriging method is more complex and is not applicable to databases with a small number of cases. IDW is easier to apply, faster, not very restrictive in terms of the number of cases per variable or in terms of variance, and usually applies to highly variable data. The IDW method was used for sediment database comprised of a reduced number of available samples (14 compared to 44 for water).

## 3. Results

*3.1. Water-Soluble Ions*

The major ions quantified by ion chromatography in water and sediment samples were $Cl^-$, $SO_4^{2-}$, $Mg^{2+}$, $Ca^{2+}$, $Na^+$, and $K^+$. For ionic species quantified in water samples, excepting $SO_4^{2-}$, higher concentrations in samples collected in October were observed (Table 1). Figure S2 in the Supplementary Materials shows the linear regressions corresponding to the ionic balance for water and sediment samples including the estimated $HCO_3^-$ contribution. For water samples (Figure S2a), compared to sediment samples (Figure S2b), an anion deficiency of 12% for samples collected in July and 18% for samples collected in October was observed. This could be associated with the presence of other organic compounds, such as amino acids and other carboxylic acids [38–41], that were not quantified in the present study.

Water hardness was calculated using the concentrations determined for $Ca^{2+}$ and $Mg^{2+}$ by ion chromatography (Table 1), and it was estimated as high as $352 \pm 18$ mg $L^{-1}$ $CaCO_3$ in July and $412 \pm 58$ mg $L^{-1}$ $CaCO_3$ in October. The values of electrical conductivity recorded in July and October were $474 \pm 121$ μS cm$^{-1}$ and $956 \pm 23$ μS cm$^{-1}$, respectively.

**Table 1.** Water-soluble ions concentrations in water (mg L$^{-1}$) samples (n = 44), pH, conductivity (S, μS cm$^{-1}$), acid-neutralizing capacity (ANC, meq L$^{-1}$), and hardness (in mg L$^{-1}$ CaCO$_3$) and water-soluble ions concentrations in sediment (μg g$^{-1}$) samples (n = 14) from Podu Iloaiei Dam Lake, north-eastern Romania. Data are presented as mean ± stdev (median) for the entire pool of analyzed samples (no flagged data).

| Species/ Parameter | Water | | Sediment | |
|---|---|---|---|---|
| | July | October | July | October |
| Cl$^-$ | 48.0 ± 5.2 (46.8) | 63.0 ± 17.7 (57.6) | 139 ± 39 (134) | 94.7 ± 64.8 (74.7) |
| SO$_4$$^{2-}$ | 234 ± 7 (235) | 181 ± 24 (179) | 1121 ± 546 (1155) | 1297 ± 484 (1170) |
| Na$^+$ | 153 ± 4 (152) | 177 ± 9 (175) | 476 ± 169 (513) | 595 ± 195 (578) |
| K$^+$ | 13.6 ± 8.3 (10.4) | 19.8 ± 15.3 (12.6) | 284 ± 98 (294) | 564 ± 133 (625) |
| Mg$^{2+}$ | 56.7 ± 1.6 (56.6) | 62.9 ± 4.1 (62.7) | 550 ± 139 (538) | 1639 ± 320 (1728) |
| Ca$^{2+}$ | 47.5 ± 6.7 (45.6) | 61.1 ± 17.8 (58.7) | 1338 ± 692 (1179) | 5041 ± 1350 (5305) |
| N-NO$_2$$^-$ | 0.40 ± 0.27 (0.40) | 0.98 ± 1.12 (0.52) | 28.0 ± 40.1 (11.4) | <1.86 |
| N-NO$_3$$^-$ | 0.56 ± 0.42 (0.47) | 1.56 ± 2.18 (0.92) | 18.8 ± 26.4 (6.2) | 7.92 ± 6.83 (8.29) |
| N-NH$_4$$^+$ | 0.65 ± 0.46 (0.55) | 2.74 ± 1.67 (2.28) | 142 ± 55 (138) | 106 ± 70 (87) |
| P-PO$_4$$^{3-}$ | <1.06 | 3.40 ± 2.20 (3.38) | 17.0 ± 10.4 (11.5) | 45.9 ± 16.7 (44.8) |
| H$_3$C$_2$O$_2$$^-$ | 32.5 ± 30.5 (24.1) | 43.0 ± 43.0 (26.3) | 394 ± 462 (230) | 223 ± 118 (202) |
| HCO$_2$$^-$ | 4.27 ± 0.74 (4.27) | 5.72 ± 6.57 (3.75) | 91.1 ± 66.3 (63.6) | 98.2 ± 47.4 (86.2) |
| C$_2$O$_4$$^{2-}$ | <7.00 | <7.00 | 64.0 ± 24.2 (60.3) | 83.1 ± 35.9 (71.2) |
| pH | 9.09 ± 0.15 (9.11) | 8.25 ± 0.07 (8.24) | | |
| S | 474 ± 121 (491) | 956 ± 23 (948) | | |
| CaCO$_3$ | 366 ± 19 (349) | 412 ± 58 (404) | | |
| ANC | 7.80 ± 0.52 (7.71) | 10.8 ± 0.8 (10.6) | | |

F$^-$, Br$^-$, Li$^+$ species were below the LoD.

The investigated inorganic ions containing nitrogen or phosphorus were NO$_2$$^-$ (N-NO$_2$$^-$), NO$_3$$^-$ (N-NO$_3$$^-$), NH$_4$$^+$ (N-NH$_4$$^+$), and PO$_4$$^{3-}$ (P-PO$_4$$^{3-}$). The N-NH$_4$$^+$ concentrations at the surface water level were 0.65 ± 0.46 mg L$^{-1}$ in July and 2.74 ± 1.67 mg L$^{-1}$ in October (Table 1). Oxalate could only be quantified in the water extracts of sediment samples, presenting lower concentrations than acetate and formate, 64.0 ± 24.2 μg g$^{-1}$ compared to 394 ± 462 μg g$^{-1}$ and 91.1 ± 66.3 μg g$^{-1}$ in July, and 83.1 ± 35.9 μg g$^{-1}$ compared to 222 ± 118 μg g$^{-1}$ and 98.2 ± 47.4 μg g$^{-1}$ in October (Table 1). C$_2$O$_4$$^{2-}$ is known to be an important component of plants [42]. It largely undergoes the process of bacterial decomposition with the formation of CO$_2$, but, in the case of hard water, C$_2$O$_4$$^{2-}$ precipitates in the form of CaC$_2$O$_4$ [43].

The water acid-neutralizing capacity (ANC) calculated with Equation (1) had the average values of 7.80 ± 0.52 meq L$^{-1}$ in July and 10.8 ± 0.8 meq L$^{-1}$ in October. These values indicate that the water is not susceptible to acidification. This aspect was confirmed by the pH measurements made in July and October, the mean values being 9.09 ± 0.15 and 8.25 ± 0.07, respectively.

$$\text{ANC} = ([\text{Ca}^{2+}] + [\text{Mg}^{2+}] + [\text{Na}^+] + [\text{K}^+]) - ([\text{SO}_4^{2-}] + [\text{NO}_3^-] + [\text{Cl}^-]) \tag{1}$$

*3.2. Water-Soluble Elements*

Table 2 shows the average concentrations of determined elements in water samples and aqueous sediment extracts for the interest sampling sessions. Higher concentrations of the quantified elements were observed for the October sampling session compared with the July sampling session.

The mean Cr concentration values were 0.019 ± 0.011 mg L$^{-1}$ and 0.064 ± 0.048 mg L$^{-1}$, in July and October, respectively. The average values of Cr concentration (Table 2) classify the lake water in quality class II according to Order 161/2006 [13]. Variations in Cr concentrations in water can be associated with adsorption/desorption and precipitation/solubilization processes [44].

The average concentrations of Pb in water samples for both sampling sessions of 0.071 ± 0.062 mg L$^{-1}$ and 0.078 ± 0.116 mg L$^{-1}$ exceeded the limits of quality class IV (according to Order 161/2006), and a concentration of 0.025 mg L$^{-1}$ was associated with

the waste water class. Measured concentrations of Ni and Mn in water samples collected in October were higher than the regulated values of 0.025 mg L$^{-1}$ and 0.30 mg L$^{-1}$, respectively, corresponding to quality class IV.

**Table 2.** Averages of water-soluble elements' concentrations in water (mg L$^{-1}$, n = 44) and sediment (µg g$^{-1}$, n = 14) samples from Podu Iloaiei Dam Lake, north-eastern Romania. Data are presented as mean ± stdev (median).

| Element | Water | | Sediment | |
|---|---|---|---|---|
| | July | October | July | October |
| Al | 1.00 ± 2.03 (0.48) | 1.35 ± 1.70 (0.68) | 96.5 ± 57.1 (79.6) | 143 ± 80 (120) |
| Ti | <0.60 | <0.60 | 8.67 ± 4.99 (7.60) | 5.36 ± 4.93 (2.94) |
| Cr | 0.019 ± 0.011 (0.019) | 0.064 ± 0.048 (0.051) | 1.21 ± 0.81 (0.79) | 1.13 ± 0.80 (0.84) |
| Mn | 0.097 ± 0.083 (0.072) | 0.33 ± 0.21 (0.34) | 20.0 ± 9.6 (18.7) | 45.4 ± 26.9 (38.2) |
| Ni | 0.12 ± 0.17 (0.09) | 1.08 ± 3.04 (0.41) | 14.4 ± 3.8 (14.2) | 17.2 ± 5.3 (17.2) |
| Zn | 0.79 ± 1.67 (0.12) | 1.88 ± 2.52 (1.27) | 15.3 ± 6.5 (17.7) | 97 ± 161 (37) |
| Ge | 0.14 ± 0.11 (0.13) | 0.19 ± 0.13 (0.15) | 19.6 ± 4.4 (20.3) | 16.8 ± 4.1 (16.9) |
| Sr | 0.54 ± 0.09 (0.52) | 0.60 ± 0.42 (0.52) | 10.6 ± 4.1 (9.4) | 40.6 ± 11.5 (39.5) |
| Ag | 0.075 ± 0.037 (0.073) | 0.13 ± 0.10 (0.10) | 0.61 ± 0.39 (0.45) | 0.95 ± 0.75 (0.63) |
| Sb | 0.006 ± 0.005 (0.005) | 0.04 ± 0.06 (0.02) | 0.26 ± 0.20 (0.18) | 0.24 ± 0.13 (0.25) |
| Ba | 0.059 ± 0.041 (0.054) | 0.16 ± 0.14 (0.12) | 22.1 ± 3.6 (21.7) | 30.3 ± 8.3 (31.5) |
| Pb | 0.071 ± 0.062 (0.062) | 0.08 ± 0.12 (0.05) | 1.93 ± 1.49 (1.52) | 1.20 ± 1.99 (0.62) |

Be, B, V, Fe, Co, Cu, Ga, Rb, Mo, Ru, Pd, Cd, Sn, Te, Ir, Tl, Bi, and U were below the LoQ.

## 4. Discussion

### 4.1. Water-Soluble Ions and Elements Related to Lake Water Typology

In both water and sediment samples the concentrations of the major ions collected in October were higher compared to July (Table 1). This was most likely the result of a significant contribution from the anthropogenic factors and less from the pedological factors or from the microbiological processes that naturally take place in the aquatic system. The Bahluet River supply water could be a source of waste for Podu Iloaiei Dam Lake because along its course fish farms, villages without sewerage, and agricultural lands are located. Differences between the two sampling seasons might also be due to the weather conditions during the collection time. In 2017, dilution of water-soluble ionic species and elements could occur in June and July due to more heavy raining events if compared to September and October. In water systems with a high-density fish population water quality can be altered by the chemicals added in order to maintain the parameters at an appropriate level. An increase in temperature, the consumption of oxygen by zooplankton, and the intensification of fodder residues' microbiological degradation processes may induce a critical decrease in dissolved oxygen. To prevent these, calcium hypochlorite (Ca(ClO)$_2$) is added to stabilize the amount of dissolved oxygen in the investigated system.

Figure 2 shows the percentage distributions of ionic species quantified in water (Figure 2a) and sediment (Figure 2b) samples and were calculated using the mean concentrations in meq L$^{-1}$ for water and µeq g$^{-1}$ (dry sediment) for sediment. From the percentage distributions presented in Figure 2, it was observed that for the water samples the major ions are present in the order Na$^+$ > HCO$_3^-$ > Mg$^{2+}$ > SO$_4^{2-}$ > Ca$^{2+}$ > Cl$^-$ > K$^+$ and for the sediment samples HCO$_3^-$ > Ca$^{2+}$ > Mg$^{2+}$ > SO$_4^{2-}$ > Na$^+$ > K$^+$ > Cl$^-$. The dominant water anion was HCO$_3^-$ followed by SO$_4^{2-}$. The high contributions of Na$^+$ (24.1%) and SO$_4^{2-}$ (14.5%) ions can be associated with the pedological factor, the investigated area having characteristic saline soils with high levels of sodium sulphate [11]. An additional source of SO$_4^{2-}$ could be represented by air pollution generated by rail transport (at ~250 m) and or by intense traffic on the road nearby.

Nitrogen and phosphorus are considered macronutrients, often limiting the development of phytoplankton. In the absence of nutrient pollution sources (fertilizers, wastewater rich in PO$_4^{3-}$, etc.), natural waters have low amounts of these species because there is a natural cycle that does not allow their accumulation in the environment. Unpolluted

natural surface waters have inorganic nitrogen concentrations of less than 0.01 mg L$^{-1}$ N-NO$_2^-$, 0.2 mg L$^{-1}$ N-NO$_3^-$, and 0.02 mg L$^{-1}$ N-NH$_4^+$ [45]. Low levels of the dissolved oxygen in the water increase the anoxic processes at the water–sediment interface. Higher concentrations of N-NH$_4^+$ (Table 1) can be determined by biodegradation processes of organic matter under anoxic conditions. NO$_2^-$ does not accumulate in the water because it is formed mainly as an intermediate in the nitrification process and has a low stability. Variations in the concentrations of P-PO$_4^{3-}$, N-NO$_2^-$, and N-NO$_3^-$ could be attributed to local processes of organic matter biodegradation. For N-NH$_4^+$, the possibility of an anthropogenic source of contamination can be considered. A seasonal source for NH$_4^+$ could be the phytoplankton degradation in early autumn, local accumulations of organic matter being possible.

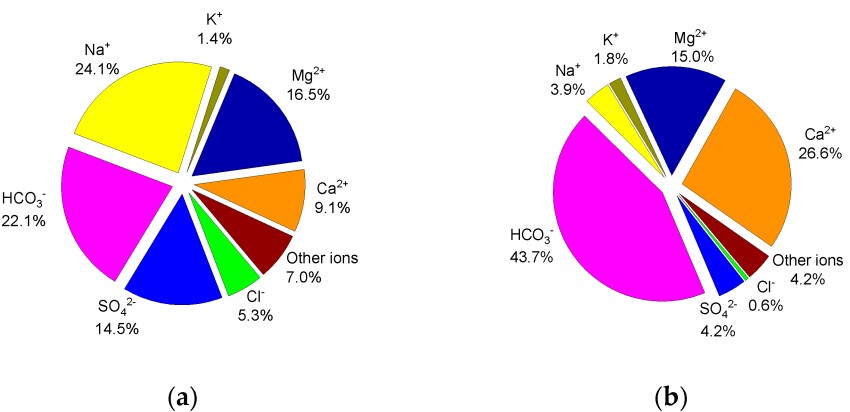

(**a**)  (**b**)

**Figure 2.** Relative concentration distribution of water-soluble ions quantified in water (**a**) and sediment (**b**) samples collected from Podu Iloaiei Dam Lake (other ions: H$_3$C$_2$O$_2^-$, HCO$_2^-$, C$_2$O$_4^{2-}$, PO$_4^{3-}$, NO$_2^-$, NO$_3^-$, NH$_4^+$).

In order to indicate the dominant typology of water, Piper diagrams were used. They allow the identification of water character by using the ratio of major ions. For water samples, the Piper diagram (Figure 3a) does not indicate the existence of a dominant water character, but the orientation towards Na$^+$ + K$^+$, HCO$_3^-$ + CO$_3^{2-}$, and SO$_4^{2-}$ can be observed. This could suggest the presence of Na$^+$, K$^+$, and SO$_4^{2-}$ sources other than the solubilization process of sediment minerals, for example, the use of fertilizers in agriculture or the discharge of wastewater from households [6]. The difference between the diagrams can also be explained by the higher solubility of the Na$^+$ and K$^+$ salts compared to those of Mg$^{2+}$ and Ca$^{2+}$. Additionally, among the major cations (Ca$^{2+}$, Mg$^{2+}$, K$^+$, Na$^+$), Ca$^{2+}$ has the lowest hydrated ion radius that enhance its adsorption on the sediment. The physicochemical properties of the soil must be studied in order to explain the ion exchange processes that take place.

For sediment (Figure 3b) samples, a dominant character of Ca$^{2+}$ (Mg$^{2+}$) HCO$_3^-$ was observed. Higher concentrations of Ca(HCO$_3$)$_2$ and CaCO$_3$ in samples collected in October compared to July could be determined by the increase in the solubility of these species when the temperature decreases, a phenomenon considered in other study [46].

Both for water (Figure 4a) and sediment (Figure 4b) samples, important variations were observed between the sampling periods for most of the elements' concentrations. These could indicate a waste accumulation in the water and sediment due to the Bahluet River contributions and fish food addition. The variations could indicate the presence of temporary pollution sources such as waste discharge and secondary pollution sources represented by the selective mobilization of metals from sediment enhanced by higher major ions concentrations in the samples collected in October compared to July.

As previously presented, the Podu Iloaiei Dam Lake water is characterized by a high content of major ions. Acosta and co-workers [47] highlighted the mobilization of Pb enhanced by an increase in CaCl$_2$, MgCl$_2$, NaCl, and Na$_2$SO$_4$ salinity. These could be

determined by the high concentrations of major ions, especially $Ca^{2+}$, $Mg^{2+}$, $Na^+$, which can lead to the mobilization of heavy metals from the sediment by the ion exchange process [2].

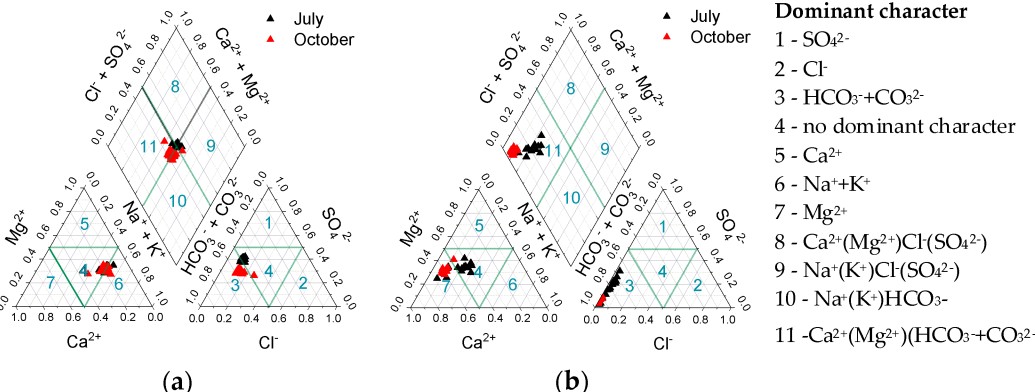

(**a**) (**b**)

**Figure 3.** Piper diagrams corresponding to water (**a**) and sediment (**b**) samples collected from Podu Iloaiei Dam Lake.

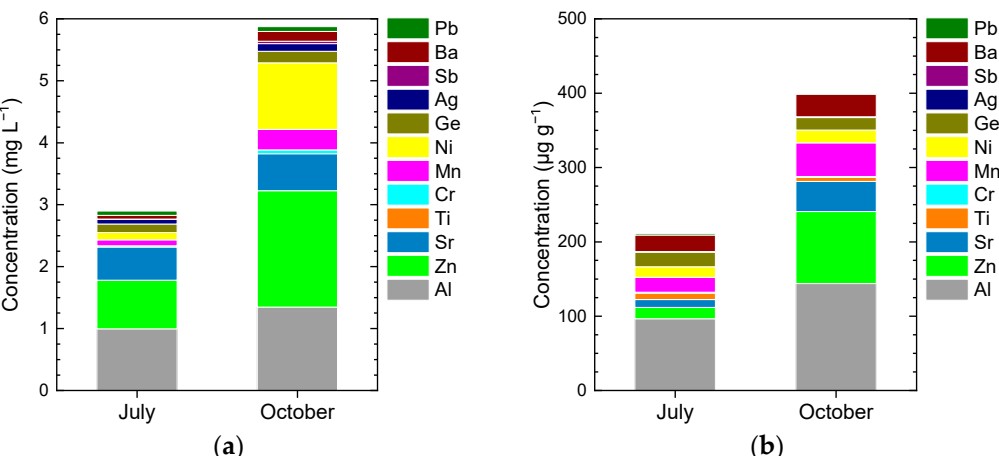

(**a**) (**b**)

**Figure 4.** The cumulative concentrations of the water-soluble elements quantified in water (**a**) and sediment (**b**) samples collected from Podu Iloaiei Dam Lake in each sampling session.

*4.2. Distribution of Chemical Parameters in the Podu Iloaiei Dam Lake*

The dendrograms obtained by the Ward method for selected species quantified in water and sediment samples collected in both July and October sampling sessions are presented in Figure 5. Prior to this analysis, the data were normalized. The distribution of chemical species within the clusters suggests that Sr, $Mg^{2+}$, $Cl^-$, $SO_4^{2-}$, $Na^+$, and $Ca^{2+}$ can be associated with the pedological factor. The association is highlighted by the Pearson's correlation coefficient values calculated and presented in Tables S4 and S5 in the Supplementary Materials. Natural contributions for these species could be soil runoff and mobilization of soluble species in water from the sediment and soil.

For the water samples (Figure 5a), a cluster including Al, Zn, Pb, $NH_4^+$, and $K^+$ was highlighted, indicating fish food administration as a possible source for these species. A common source for $NH_4^+$ and $K^+$ could be the use of fertilizers in agriculture and the bacterial degradation of plant residues rich in organic compounds with N and $K^+$. The Sb, Ti, Ge, Ba, Mn, Cr, Zn, and Ag determined in sediment were grouped in one cluster, suggesting that the main contribution for these elements could be attributed to the pedological factor. A strong association between Pb and Al in sediment could indicate their anthropogenic origin. Additionally, a statistically significant correlation factor of 0.77 ($p < 0.001$) (Table S5 in the Supplementary Materials) between these elements was observed.

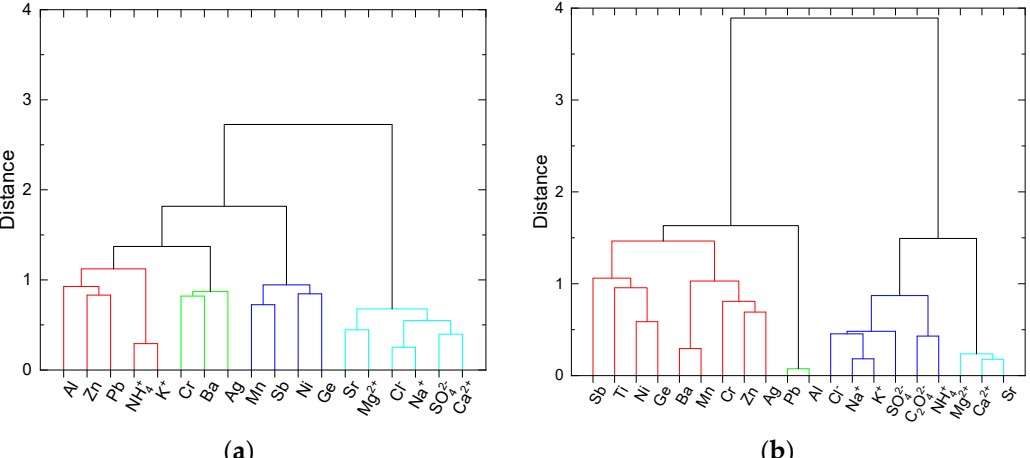

**Figure 5.** Hierarchical dendrograms showing clustering of water-soluble ions and elements in water (**a**) and sediment (**b**) samples collected in July and October sampling sessions.

Geospatial distributions in the water samples of the pH and conductivity parameters, beside those of water-soluble ions as $Mg^{2+}$, $Ca^{2+}$, $NH_4^+$, $Na^+$, $SO_4^{2-}$, $Cl^-$, and $NO_3^-$, are shown in Figure 6. Figure S3 of the Supplementary Materials presents the geospatial distributions of $K^+$, $NO_2^-$, $H_3C_2O_2^-$, and $HCO_2^-$ concentrations in both water and sediment samples. There was a clear discrimination between areas with two different particularities in terms of potential existent acidity and alkaline driven species. From Figure 6 it is clearly observed that water-soluble $Mg^{2+}$, $Ca^{2+}$, and $NH_4^+$ are major ionic species controlling alkalinity. While over the July session, the $NH_4^+$ species seemed to play an important role in the area where the water depth was lower, in the October session, it was obvious that the importance of this ion in controlling water pH might enhance in the feeding area. The values of the pH parameter measured for both July and October sessions allow us to suggest the potential transition of $NH_4^+$ to the toxic and volatile $NH_3$, which might occur in the investigated water lake.

In water samples, several common point sources are easily identified in the geospatial distributions of $Ca^{2+}$, $Cl^-$, and $NO_3^-$ (Figure 6). Areas with higher conductivity (Figure 6) correspond to areas with higher $Mg^{2+}$, $NH_4^+$ (Figure 6), and $K^+$ (Figure S3 in the Supplementary Materials) concentrations.

Figure 7 shows the distribution of $Mg^{2+}$, $Ca^{2+}$, $NH_4^+$, $Na^+$, $SO_4^{2-}$, $Cl^-$, $NO_3^-$, and $C_2O_4^{2-}$ in sediment samples. Similar geospatial distributions of $Mg^{2+}$, $Ca^{2+}$, $Na^+$, $SO_4^{2-}$ (Figure 6), and $K^+$ (Figure S3 in the Supplementary Materials) concentrations were observed. $C_2O_4^{2-}$ was quantified only in sediment samples with higher concentration values in the feeding area (Figure 6). Since the investigated water lake is assigned as hard water, $C_2O_4^{2-}$ is most probably present as a precipitate in the form of $CaC_2O_4$.

The spatial distributions for the elements Sr, Cr, Pb, Mn, Zn, Ba, Ni, and Sb in the water and sediment samples collected from Podu Iloaiei Dam Lake are presented in Figures 8 and 9, while Figure S4 of the Supplementary Materials shows the distributions of Al, Ge, and Ag. The Sr concentration variations in the sediment (Figure 9) present a similar profile with $Mg^{2+}$, $Ca^{2+}$, $Na^+$, and $SO_4^{2-}$ (Figure 6). In the south-eastern part of the lake, at the lateral end of the dam, in both sampling sessions, a spot corresponding to high concentrations for Cr, Pb, Mn, Zn, Ba, Ni (Figures 8 and 9), and Al (Figure S4 in the Supplementary Materials) was observed, indicating the existence of a punctual and persistent source of pollution. Geospatial distributions describe different areas with high and low concentrations for Cr and Pb in water (Figure 6) and sediment (Figure 7) samples. These highlight different sources or mobilization processes for these elements. For water samples collected in October, an increase in $Mg^{2+}$ (Figure 6), $K^+$ (Figure S3 in the Supplementary Materials), Mn, and Zn (Figure 8) concentrations was observed in the feeding area, possibly generated by the fish food addition.

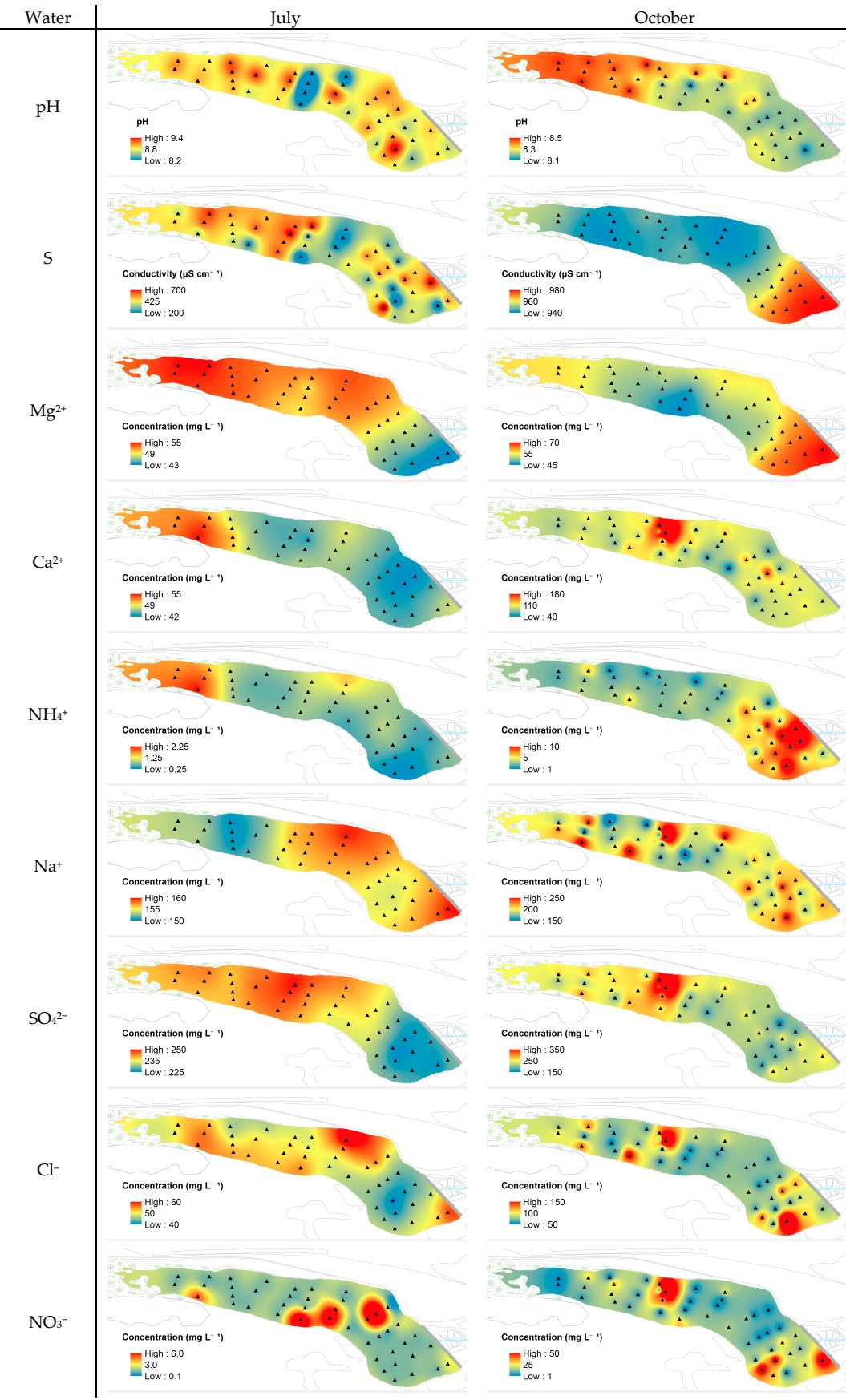

**Figure 6.** Spatial distribution of pH, conductivity (S), $Mg^{2+}$, $Ca^{2+}$, $NH_4^+$, $Na^+$, $SO_4^{2-}$, $Cl^-$, and $NO_3^-$ in water of Podu Iloaiei Dam Lake for July and October sampling sessions.

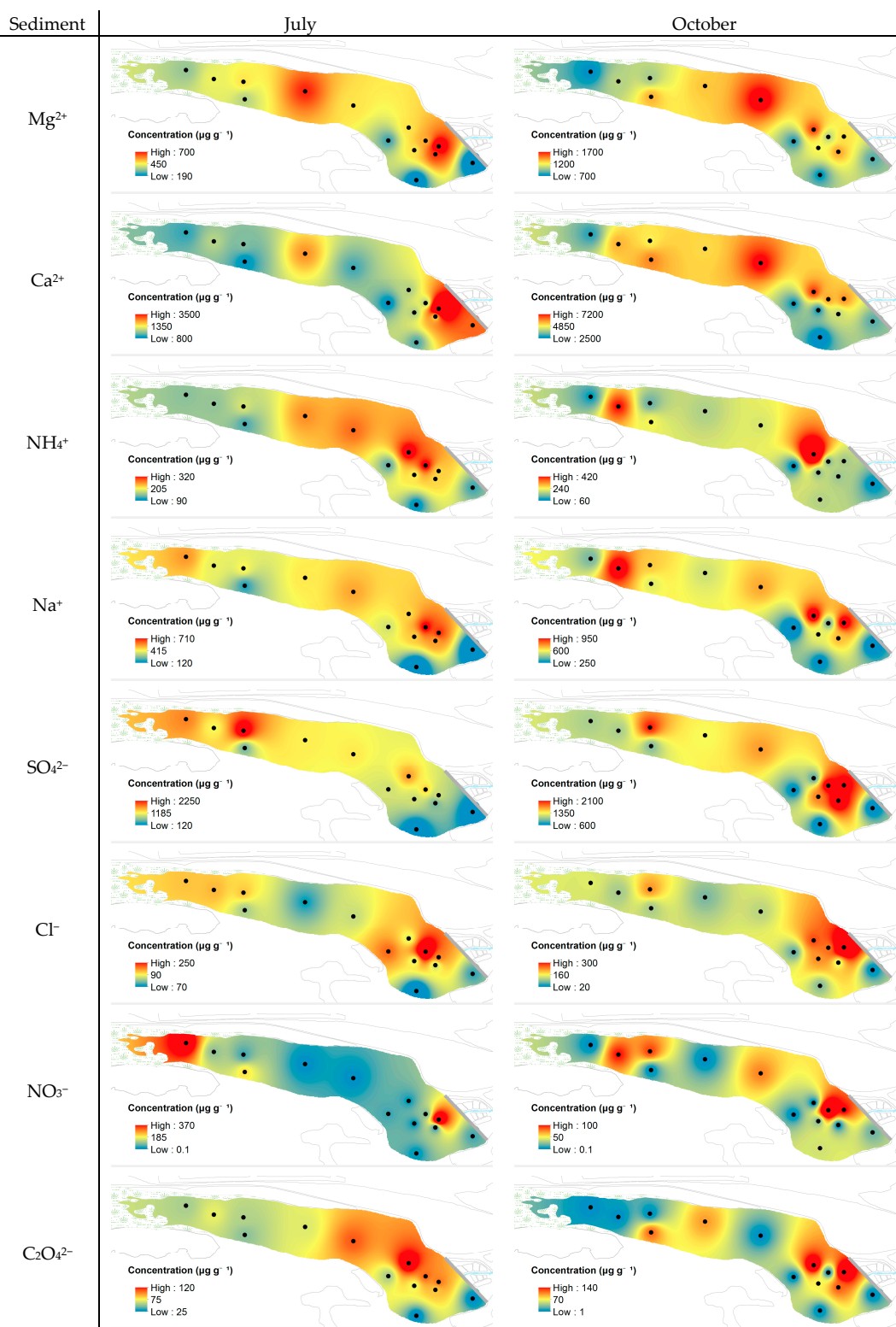

**Figure 7.** Spatial distribution of $Mg^{2+}$, $Ca^{2+}$, $NH_4^+$, $Na^+$, $SO_4^{2-}$, $Cl^-$, $NO_3^-$, and $C_2O_4^{2-}$ in sediment of Podu Iloaiei Dam Lake for July and October sampling sessions.

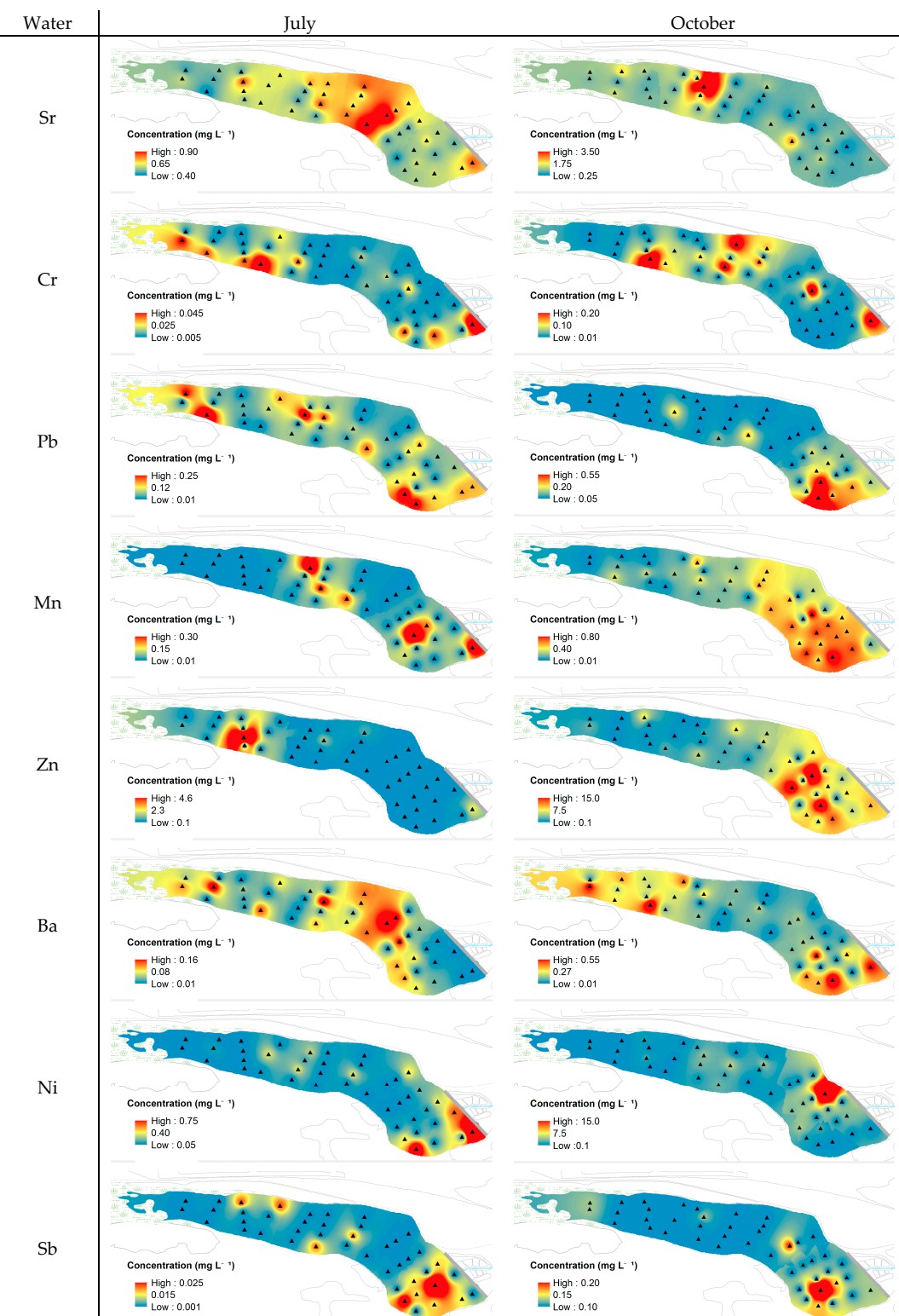

**Figure 8.** Spatial distribution of Sr, Cr, Pb, Mn, Zn, Ba, Ni, and Sb in water of Podu Iloaiei Dam Lake for July and October sampling sessions.

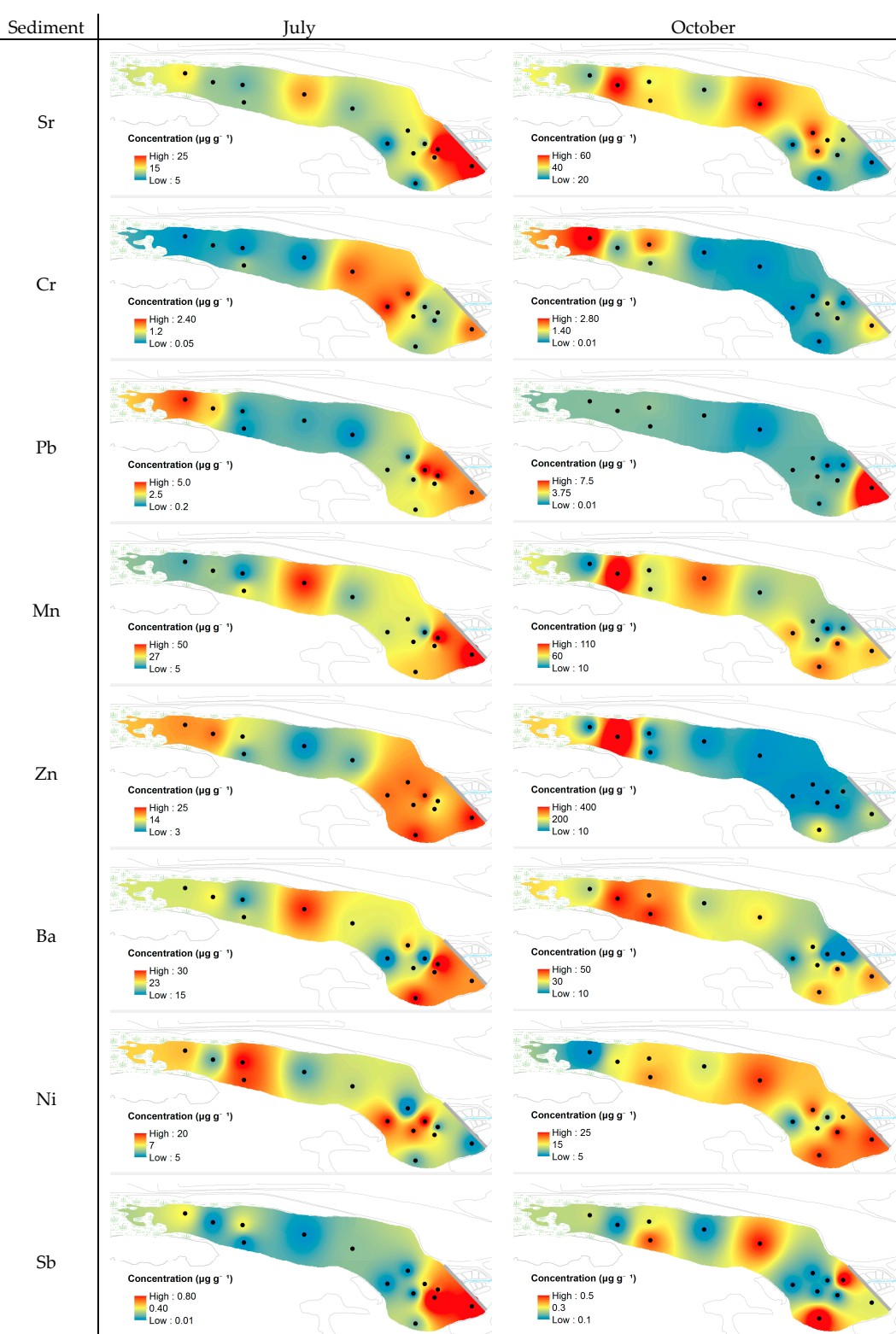

**Figure 9.** Spatial distribution of Sr, Cr, Pb, Mn, Zn, Ba, Ni, and Sb in sediment of Podu Iloaiei Dam Lake for July and October sampling sessions.

The Kolmogorov–Smirnov test applied to the database indicated significant different distributions of Cr, Mn, Ni, Zn, Sb, Ba, $Cl^-$, $SO_4^{2-}$, $Na^+$, $NH_4^+$, $K^+$, $Mg^{2+}$, and $Ca^{2+}$ concentrations. For Al, Ge, Sr, Ag, and Pb present in water samples collected in different sampling sessions from Podu Iloaiei Dam Lake the differences were not significant. Additionally,

significant different concentrations were observed for Al, Ti, Mn, Sr, Ag, Ba, Pb, $Cl^-$, $K^+$, $Mg^{2+}$, and $Ca^{2+}$, while insignificant differences were noticed for Cr, Ni, Zn, Ge, Ag, Sb, $SO_4^{2-}$, $C_2O_4^{2-}$, $Na^+$, and $NH_4^+$ in sediment samples collected in both seasons from the investigated water.

*4.3. Suitability of the Podu Iloaei Lake for Aquaculture Purposes and Potential Practical Implications*

Water quality is one of the main concerns in aquaculture [9]. Changes in the chemical or physical parameters can cause negative effects on the growth of the aquatic organisms, their physiological state, or even mortality, with major losses in production.

The physicochemical parameters measured in small-scale fish production systems are pH, conductivity, and dissolved oxygen. Elements' presence or other ions in water sources are not generally measured. Besides the water source, the quality of the effluent water is highly important when considering environmental aspects of fish production. Aquaculture's increase over the last years could have an important impact on the environment and natural water resources. Some concerns include: water pollution resulting from pond effluents (i.e., salinization of water by effluents, seepage, and sediment), excessive use of ground water and other freshwater sources to increase the water level in ponds, the spread of aquatic animal diseases from cultured organisms to native populations, and negative effects on biodiversity caused by the escape of non-native species introduced for aquaculture [9].

The lake water (Podu Iloaiei Dam Lake) investigated in the present study had a hardness higher than 180 mg $L^{-1}$ $CaCO_3$, and therefore it was considered a very hard water. Waters with a hardness higher than 300 mg $L^{-1}$ $CaCO_3$ do not provide optimal conditions for high fish productivity [7]. It is recommended that the pH of the water be in the 7.0–8.5 range because fish blood has an average pH of 7.4 [1]. The pH values measured in the present work over both the July and October sampling sessions allowed us to suggest that precautions be taken by the Podu Iloaiei Dam Lake administrator in order to prevent the potential negative influence of increased alkalinity on fish productivity.

Moreover, attention should be given to the water status in full compliance with the requirements derived from the National Agency for Environmental Protection Report on the state of the environment in the Iasi County [25]. The report clearly shows that from six dam lakes located in the Prut basin, five lakes did not meet the moderate ecologic potential (Class III) [25]. However, while in the report from 2018, the Podu Iloaiei Dam Lake was classified as moderate with no traceable measurements for specific chemical parameters [25], in the report covering measurements from 1997 to 2004, the lake water was included either in class IV (pH, $CBO_5$, CCO-Mn, $NH_4^+$, $Mg^{2+}$) or in class III ($NO_2^-$, $NO_3^-$, $PO_4^{3-}$) [14] of quality. Considering the total nitrogen and phosphorus levels reported in 2014 [15] the water lake was assigned as hypertrophic.

In the present study, according to the classification of waters in the quality classes, used to establish the ecological status of water bodies [13], the Podu Iloaiei Dam Lake water falls in water quality class III based on the $SO_4^{2-}$, $Na^+$, $Mg^{2+}$ and nutrient contents. This suggests that Podu Iloaiei Dam Lake can be suitable for aquaculture, but it is not recommended to breed fish species sensitive to high hardness and high ionic strength because they can suffer imbalances at the osmoregulatory system level. The productivity could also be reduced due to the difficulty of nutrient adsorption in such conditions [7]. Higher concentrations of water-soluble ions in October could indicate an increase in dissolved solids, which has the effect of decreasing the gases' solubility in water, which can also negatively affect the osmotic balance of fish [48].

Considering the trophicity classes proposed by Dodds and co-researchers [49], with regard to nutrient concentration limits, the Podu Iloaiei Dam Lake water has an oligotrophic character in July, while, in October, the nutrient level corresponds to the eutrophic waters. Factors that could cause this variation are the discharge of waste waters, the administration of fertilizers in the agricultural area, the administration of fish food, and low rainfall that implied reduced discharge in order to keep the optimal level of water.

From the investigated elements, Pb seems to be highly influenced by the anthropogenic factor, considering that the average concentration of Pb in water samples exceeded the limits for the quality class IV associated with the waste water category.

Overall, the Podu Iloaiei Dam Lake seems to have chemical characteristics of a system controlled by both natural and anthropogenic origin sources. A complete database in terms of common chemical parameters; chemical pollutants; parameters representative for aquaculture activities, such as dissolved oxygen, biological oxygen demand, and chemical oxygen demand; and potential chemical processing of the specific water system would be of great benefit for the administration of such water resources in order to develop appropriate planning and lake exploitation. Moreover, such information might be used by experts in the field of developing lake management policies for a sustainable exploitation of various aquatic systems.

## 5. Conclusions

Water and sediment samples from Podu Iloaiei Dam Lake located on the Bahluet River from the Bahlui hydrographic basin, north-eastern Romania, were investigated in order to determine their chemical composition in terms of the water-soluble constituents.

Anthropogenic and natural (pedologic) related sources seem to control the chemical composition of the water and sediment samples in the area. Experimental evidence obtained within the present work allowed us to infer that the Podu Iloaiei Dam Lake water was not susceptible to acidification. The trophicity of the investigated lake water seemed to present seasonal variability, changing from oligotrophic during warm periods to eutrophic over cold periods. Moreover, the Podu Iloaiei Dam Lake water traces the characteristics of a potential saline soil existent in the nearby region. The abundance of water-soluble chemical species suggests not only the solubilization process of sediment minerals but also the use of fertilizers in agriculture as important sources in the water body of interest. Uncontrolled or accidental waste discharge and secondary pollution sources might have contributions to the water characteristics.

The complex interplay between the water body's main characteristics, meteorological factors, and anthropogenic activities in the area control the abundances and the association of chemical species present in the water body. The results of the present study allow us to suggest that the Podu Iloaiei Dam Lake is suitable for aquaculture activities, but it is not recommended to breed fish species sensitive to high hardness and high ionic strength because they can suffer imbalances at the osmoregulatory system level.

The present study reports on the investigation of chemical parameters that are not routinely monitored at the European scale but which might have relevance for further studies concerning photochemical processing or when investigating health issues related to the consumption of aquaculture products exposed to potential toxics. However, due to various sources of water pollution, further studies might also evaluate and consider other important parameters such as dissolved oxygen, biological oxygen demand, and chemical oxygen demand as indicators for pollution. Moreover, further research could also involve measurements of physicochemical parameters in other water bodies (e.g., Bahluet River) in the region of interest. The evaluation of different water chemical treatments' impact on fish farming productivity and water quality in the Bahlui hydrographic basin might also be of interest.

**Supplementary Materials:** The following are available online at https://www.mdpi.com/article/10.3390/w13172395/s1, Table S1: Classification of waters in the quality classes in order to establish the ecological status of water bodies (extracted from Order no. 161/2006), Table S2: Limits of detection (LoD) and limits of quantification (LoQ) for ion chromatography determined water-soluble ions in the water and sediment samples, Table S3: Limits of detection (LoD) and limits of quantification (LoQ) for inductively coupled plasma mass spectrometry determined elements in the water and sediment samples, Table S4: Pearson's correlation coefficient of chemical species, pH, and conductivity (S) determined in water samples collected from Podu Iloaiei Dam Lake, Table S5: Pearson's correlation coefficient of chemical species quantified in sediment samples collected from Podu Iloaiei Dam Lake,

Figure S1: Linear regression between ($\sum$cations $-$ $\sum$identified anions) and ([$Mg^{2+}$] + [$Ca^{2+}$]) for water samples (a) and sediment (b) samples in both sampling sessions, Figure S2: Linear regression for the ionic balance evaluation of water (a) and sediment (b) samples (including the $HCO^{3-}$ estimation), Figure S3: Spatial distribution of $K^+$, $NO^{2-}$, $H_3C_2O^{2-}$, and $HCO^{2-}$ in water and sediment of Podu Iloaiei Dam Lake, Figure S4: Spatial distribution of water-soluble Al, Ge, Ag, and Sb in water and sediment of Podu Iloaiei Dam Lake.

**Author Contributions:** Conceptualization, C.A. (Cornelia Amarandei), S.-M.C.-M., R.-I.O., and C.A. (Cecilia Arsene); methodology, C.A. (Cornelia Amarandei), R.-I.O., and C.A. (Cecilia Arsene); methods validation, C.A. (Cornelia Amarandei), A.-G.N., and L.-V.S.; formal analysis, C.A. (Cornelia Amarandei), A.-G.N., L.-V.S., S.-M.C.-M., R.-I.O., and C.A. (Cecilia Arsene); investigation, C.A. (Cornelia Amarandei), A.-G.N., L.-V.S., S.-M.C.-M., R.-I.O., and C.A. (Cecilia Arsene); resources, S.-M.C.-M., R.-I.O., and C.A. (Cecilia Arsene); data curation, C.A. (Cornelia Amarandei), A.-G.N., L.-V.S., S.-M.C.-M., R.-I.O., and C.A. (Cecilia Arsene); writing—original draft preparation, C.A. (Cornelia Amarandei); writing—review and editing, A.-G.N., L.-V.S., S.-M.C.-M., R.-I.O., and C.A. (Cecilia Arsene); supervision, C.A. (Cecilia Arsene). All authors have read and agreed to the published version of the manuscript.

**Funding:** This research received no external funding.

**Data Availability Statement:** Data are available upon request to the contact author Cecilia Arsene (carsene@uaic.ro).

**Acknowledgments:** The authors express their gratitude to the Administration of the Podu Iloaiei Dam Lake for facilitating access in the area for sampling purposes. Acknowledgment is given by S.-M.C.-M., R.-I.O., and C.A. (Cecilia Arsene) to the infrastructure support from the Operational Program Competitiveness 2014–2020, Axis 1, under POC/448/1/1 research infrastructure projects for public R&D institutions/Sections F 2018, through the Research Center with Integrated Techniques for Atmospheric Aerosol Investigation in Romania (RECENT AIR) project, under grant agreement MySMIS no. 127324. Acknowledgments are given to the CNFIS-FDI-2021-0501 project as well.

**Conflicts of Interest:** The authors declare no conflict of interest.

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
