# Peer review of "Assessment of Surface Water Quality in the Podu Iloaiei Dam Lake (North-Eastern Romania): Potential Implications for Aquaculture Activities in the Area"

_water, doi:10.3390/w13172395_

Round 1

Reviewer 1 Report

The reviewed article focuses on the chemical characteristics of the water and sediments samples derived from the Podu Iloaiei Dam Lake. First of all, the topic  as the obtained results are in line in the scope of the journal and may be of  interest to the reads. However crucial improvements have to be made prior to the final publication. The manuscript in its current form looks more like simple report from two measurements campaigns, rather than fully professional article. Please check specified comments about issues that need some special attention.

Specific comments.

  1. Abstract. The abstract needs to be refined. Moreover, key findings should be presented in refined manner. The abstract immediately jumps into the study results, without first explaining the set-up of/reasoning behind the study. Readers would benefit from a larger portion of the abstract spent on this background info and a more concise summary of the results. There is no need to list all of the performed measurements - interested reader will go  to M&M section. Show only some examples from the particular group. Presented findings seem to be too general, expose the most crucial findings.
  2. Introduction. This section must be completed with a better explanation of the purpose of the research undertaken. Moreover, a broader context of the previous similar studies has to be presented. What has so far been analyzed in the framework of similar studies in the country of the authors and/or other countries from the EU?
  3. Material and methods. Please marked on the map in Figure 1, land use around the lake. This may help to better understand point sources of the particular contamination a) What where criteria of the sampling points selection?b) Why water sampling was limited to the single depth? c) I wonder if any repetitions of the samples has been made? d) Why there are no measurements of the COD or TOC, i.e. organic compounds concentrations? This are the crucial parameters required to understand trophic state of the water. e) What about total suspended solids? They may affect distribution of the particular ions between water column and sediments.
  4. Results. Please provide wider explanation how classification to water quality class has been made. Readers may be interested in how this procedure similar or differs from other European countries. It will be good point to present water quality requirements in Romania law in the supplementary section to better motivated choice of analysis performed in frame of this article. Obtained results require better discussion with the reports from similar studies and some governmental summaries about water quality to give wider context of the performed analysis.
  5. Conclusions. The conclusions section needs to be trimmed and refined. In conclusion section there is no need to repeat general information but emphasize the most  pronounced findings.

Reviewer 2 Report

This is an interesting study on surface water quality. The research methods are correct, but some revisions are needed to improve the scientific soundness of this study.

  1. The Introduction section: please justify why the Podu Iloaiei water reservoir is important to investigate to increase the scientific justification for the international readers.
  2. Material and methods: lines 163-168 refer to the results rather than methods, please consider revisions. 
  3. The results section: please consider renaming paragraph 4.3  - the headline should clearly refer to the findings described in the text (e.g. distributions of....)
  4. Please consider adding 2-3 sentences on the practical implications of this study and further research needs. 
  5. The conclusions section is to look. Please do not repeat findings and methods but only focus on the most important conclusions and "take-home-message" based on the obtained findings.

Reviewer 3 Report

General comment:

The present study is interesting and potentially could contribute to the research field, however, there are some concerns that require be addressed to clarity and improve the present version.

Specific comments:

  1. In the abstract, the use of warm and cold sessions seems not appropriate.
  2. In the introduction, a brief discussion on water quality and fish farming should be incorporated.
  3. In the results, (page 5, line-225), the Authors mentioned only results. It would be better to write results and discussion. There are the same subsections (3.1. Water-Soluble Ions; 3.2. Water-Soluble Elements; 4.1. Water-Soluble Ions; 4.2. Water-Soluble Elements) two times appear in the results. Why?
  4. On page 10, line 353, “4.3. Statistical Analysis of the Data” is not appropriate. Authors should choose alternative and appropriate heading.
  5. Discussion should be more focused with appropriate references.
  6. The conclusion should be more concise by describing the aims and major findings of the study. Major limitations and opportunities to inform future research are not addressed properly.
  7. The meaning of some sentences are not easily understandable (i.e., L-30-34; L-37)
  8. Dissolved oxygen (DO), Biological oxygen demand (BOD), Chemical oxygen demand (COD), Total dissolved solids (TDS), pH are important parameters for water quality assessment, why authors did not consider these parameters? Is there any explanation?

Round 2

Reviewer 1 Report

Third version of the manuscript is slightly improved in relation to the first version. All of my suggestions have been addressed.

Reviewer 2 Report

The authors addressed all the comments.

Reviewer 3 Report

The revised manuscript seems good, however, authors should more careful and review the abstract and conclusion section. Please provide gist message concisely and rewrite by removing less important sentences from the abstract and conclusion.
